# The Diagnostic Performance of Multi-Detector Computed Tomography (MDCT) in Depiction of Acute Spondylodiscitis in an Emergency Department

Alberto Negro [1,*], Francesco Somma [1], Mario Tortora [2], Marina Lugarà [3], Stefania Tamburrini [4], Maria Gabriella Coppola [3], Valeria Piscitelli [1], Fabrizio Fasano [1], Carmine Sicignano [1], Ottavia Vargas [1], Gianvito Pace [1], Mariarosaria Giardiello [1], Michele Iannuzzi [5], Gabriella Toro [4], Fiore De Simone [4], Marco Catalano [4], Roberto Carbone [4], Concetta Rocco [4], Pietro Paolo Saturnino [4], Luigi Della Gatta [6], Alessandro Villa [7], Fabio Tortora [2], Laura Gemini [2], Ferdinando Caranci [8] and Vincenzo D'Agostino [1]

1 Department of Neuroradiology, Ospedale del Mare, Via Enrico Russo, 80147 Naples, Italy; fra1585@hotmail.it (F.S.); valeria.piscitelli@gmail.com (V.P.); fabriziodoc@gmail.com (F.F.); carmine.sicignano@gmail.com (C.S.); ottaviavargas@gmail.com (O.V.); dr.gianvito.pace@gmail.com (G.P.); mariarosariagiardiello@gmail.com (M.G.); vincenzo-dagostino@libero.it (V.D.)
2 Department of Advanced Biomedical Sciences, Federico II University Naples, 80131 Naples, Italy; mario.tortora@ymail.com (M.T.); fabio.tortora@unina.it (F.T.); laura.gemini93@gmail.com (L.G.)
3 Department of Internal Medicine, Ospedale del Mare, Via Enrico Russo, 80147 Naples, Italy; marinalugara82@gmail.com (M.L.); gabry.cop@libero.it (M.G.C.)
4 Department of Radiology, Ospedale del Mare, Via Enrico Russo, 80147 Naples, Italy; tamburrinistefania@gmail.com (S.T.); gabriella.toro@tiscali.it (G.T.); desimonef@libero.it (F.D.S.); marco26catalano@yahoo.it (M.C.); robcarbone@alice.it (R.C.); dottimmarocco@gmail.com (C.R.); pietropsaturnino@libero.it (P.P.S.)
5 Department of Anesthesia and Intensive Care, Ospedale del Mare, Via Enrico Russo, 80147 Naples, Italy; michele.ianuzzi74@gmail.com
6 Department of Neuoradiology, AORN A. Cardarelli, Via Antonio Cardarelli, 80131 Naples, Italy; luigi.dellagatta@hotmail.it
7 Department of Neurorsurgery, Ospedale del Mare, Via Enrico Russo, 80147 Naples, Italy; alessandrovilla83@gmail.com
8 Department of Precision Medicine, University of Campania Luigi Vanvitelli, Via de Crecchio, 80138 Naples, Italy; ferdinando.caranci@unicampania.it
* Correspondence: alberto.negro@hotmail.it

**Abstract:** Background: The diagnosis of acute spondylodiscitis can be very difficult because clinical onset symptoms are highly variable. The reference examination is MRI, but very often the first diagnostic investigation performed is CT, given its high availability in the acute setting. CT allows rapid evaluation of other alternative diagnoses (e.g., fractures), but scarce literature is available to evaluate the accuracy of CT, and in particular of multi-detector computed tomography (MDCT), in the diagnosis of suspected spondylodiscitis. The aim of our study was to establish MDCT accuracy and how this diagnostic method could help doctors in the depiction of acute spondylodiscitis in an emergency situation by comparing the diagnostic performance of MDCT with MRI, which is the gold standard. Methods: We searched our radiological archive for all MRI examinations of patients who had been studied for a suspicion of acute spondylodiscitis in the period between January 2017 and January 2021 ($n$ = 162). We included only patients who had undergone MDCT examination prior to MRI examination ($n$ = 25). The overall diagnostic value of MDCT was estimated, using MRI as the gold standard. In particular, the aim of our study was to clarify the effectiveness of CT in radiological cases that require immediate intervention (stage of complications). Therefore, the radiologist, faced with a negative CT finding, can suggest an elective (not urgent) MRI with relative serenity and without therapeutic delays. Results: MDCT allowed identification of the presence of acute spondylodiscitis in 13 of 25 patients. Specificity and positive predictive value were 100% for MDCT, while sensitivity and negative predictive value were 68% and 50%, respectively, achieving an overall accuracy of 76%. In addition, MDCT allowed the identification of paravertebral abscesses (92%), fairly pathognomonic lesions of spondylodiscitis pathology. Conclusions: The MDCT allows identification of the presence of

---

acute spondylodiscitis in the Emergency Department (ED) with a satisfactory accuracy. In the case of a positive CT examination, this allows therapy to be started immediately and reduces complications. However, we suggest performing an elective MRI examination in negative cases in which pathological findings are hard to diagnose with CT alone.

**Keywords:** spondylodiscitis; multi-detector computer tomography (MDCT); magnetic resonance imaging (MRI)

## 1. Introduction

Spondylodiscitis is usually used in the medical field to define osteomyelitis of the spine associated with adjacent disc inflammation. It is not a common pathology even if its incidence is increasing (about 2.4/100,000). An early diagnosis is essential to prevent serious related sequelae [1–3]. It is provoked by an infection that could originate from a distant focus, spreading through the bloodstream, or from a contiguous focus such as nearby soft tissues, most commonly iatrogenic after spinal surgery [4]. Susceptible patients are elderly, immunosuppressed, nephropathic, and drug addicts who use intravenous injectable drugs [5]. The clinical symptomatology is generally non-specific, characterized by back or abdominal pain, fever, and fatigue, and is often masked by previous self-medication. Inflammatory blood values could be elevated, but blood cultures are often negative [5]. In light of this, diagnosis is difficult and sometimes delayed even for months [4–6]. MRI, with sensitivity and specificity higher than 95%, represents the radiological gold standard, allowing a precise evaluation of the pathology [5,7–9]. However, especially in emergency situations, the first diagnostic investigation is CT, which is faster, easily available, and can also be performed in claustrophobic patients or in presence of incompatible MRI devices. It allows the identification of spine pathologies such as fractures, which could be a plausible differential diagnosis in a patient with non-specific clinical manifestations [10–13]. The number of published studies on the diagnostic accuracy of CT in the depiction of suspected spondylodiscitis is scant, and most of these studies have been conducted with older generation CT scanners [14,15]. Now, thin-slice multi-detector CT (MDCT) scanners allow images characterized by high spatial and contrast resolution and with high detail multiplanes reconstructions to be obtained. [16–18]. The aim of our study was to establish MDCT accuracy and to demonstrate how this diagnostic method could help doctors in the depiction of acute spondylodiscitis in an emergency situation by comparing the diagnostic performance of MDCT to MRI, the gold standard.

## 2. Materials and Methods

### 2.1. Patients

In our retrospective observational study, we searched in the Digital Archive for all MRI exams of suspected spondylodiscitis in the period between January 2017 and January 2021 and identified 162 patients (average age: 63.18 ± 4.47 years; M/F = 87/75). Of these, we include only patients who had undergone MDCT examination prior to MRI examination ($n$ = 25; 15.44%), excluding patients without MDCT performed before MRI exam ($n$ = 137; 84.56%). The overall diagnostic value of MDCT was estimated, using MRI as the gold standard. In the CONSORT flow diagram, we report the study design and the population included in the study (Figure 1).

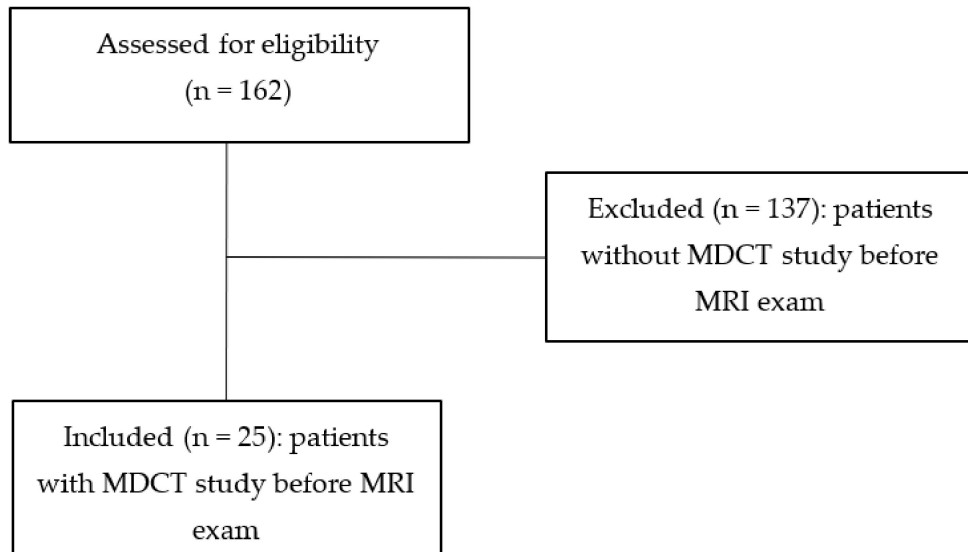

**Figure 1.** CONSORT flow diagram of selection of patients under study.

In Table 1, the demographic data of the 25 included patients (medium age: 64.72; M/F = 15/10) are reported.

**Table 1.** Demographic data of the subjects included in the study.

| Patients | Sex | Age |
|:---:|:---:|:---:|
| 1 | M | 60 |
| 2 | F | 65 |
| 3 | M | 61 |
| 4 | M | 63 |
| 5 | M | 63 |
| 6 | F | 70 |
| 7 | M | 57 |
| 8 | M | 58 |
| 9 | M | 57 |
| 10 | F | 65 |
| 11 | M | 63 |
| 12 | F | 70 |
| 13 | M | 61 |
| 14 | M | 69 |
| 15 | F | 64 |
| 16 | M | 60 |
| 17 | F | 71 |
| 18 | M | 67 |
| 19 | M | 65 |
| 20 | F | 70 |
| 21 | F | 64 |
| 22 | M | 65 |

**Table 1.** *Cont.*

| Patients | Sex | Age |
|:---:|:---:|:---:|
| 23 | F | 71 |
| 24 | F | 70 |
| 25 | M | 69 |

## 2.2. MR Imaging

MRI exams were conducted with a 1.5T scanner (Amira, Siemens Medical Solutions, Erlangen, Germany) using a multi-channel receive-only surface coil. Each MRI study was based on performing T1-weighted spin-echo (SE) pulse sequences, T2 turbo spin-echo (TSE), and short T2-weighted pulse sequences (STIR), with tau inversion recovery sequences in sagittal planes and TSE T2-weighted pulse sequences in axial planes before administering gadolinium-based contrast agent (TR/TE: 400–465/9 for T1-weighted sequences and TR/TE 2000–3500/100 ms for T2-weighted sequences). A total of 24 examinations (96.0%) were completed with enhanced T1-weighted Dixon pulse sequences and enhanced FAT SAT SE T1-weighted pulse sequences in sagittal and axial planes. A slice thickness of 4 mm and a field of view of 340 × 340 mm were used for all sequences. Only 1 (4.0%) examination was not completed with intravenous injection of paramagnetic contrast medium due to patient renal dysfunction (glomerular filtration rate < 30mg/ dL).

## 2.3. MDCT Imaging

Computer tomography examinations were performed using a multi-slice CT system (Aquilion 64, Toshiba Medical Systems, Fukuoka, Japan). A total of 10 patients (40.0%) underwent CT study without intravenous contrast medium, (non-enhanced CT (NECT)), while the remaining 15 (60%) patients were evaluated with CT examination before and after intravenous administration of contrast medium (enhanced CT (CECT)). All MDCT examinations were performed by adequate volumetric spiral acquisition with a slice thickness of 2 mm, using a bone and soft tissue kernel, a 512 × 512 matrix, and a 40 × 40 cm FOV.

## 2.4. Image Analyses

The image analysis was conducted by our radiologists (with a certified experience of 5–30 years). In our evaluation, we use the following established criteria for acute spondylodiscitis:

For MRI: vertebral and disc edema, inflammatory tissue in the paravertebral or epidural space, erosive bone changes, and pathological enhancement of the vertebrae and intervertebral discs after intravenous injection of gadolinium paramagnetic contrast medium [19].

For MDCT: erosive changes affecting the vertebrae at the level of the bodies and end plates, and presence of inflammatory tissue affecting the paravertebral or epidural space with abscess formation [19,20].

Diagnosis was further confirmed by positive blood cultures and clinical follow-up or by positive microbiological exams obtained by vertebral biopsy under CT or X-ray guidance.

## 2.5. Statistical Analysis

The evaluation of the diagnostic sensitivity, specificity, and accuracy of the MDCT and MRI was performed with appropriate contingency tables using Statistical Package for Social Science package (SPSS Inc., Armonk, NY, USA, v.17.0). In addition, we presented as a descriptive statistic the frequency of lesions and radiological features depicted by either MDCT or MRI.

### 3. Results

We identified the presence of acute spondylodiscitis in 19/25 cases (76%). A causative bacterial pathogen was documented in each case. More than half (11/19; 57.9%) were *S. aureus*—among which 5 (45.5%) showed methicillin resistance; 4/19 (21%) were represented by Streptococcus species, 3/19 (15.8%) by Escherichia coli, and 1/19 (5.3%) by *Candida* sp.

Of these 19 patients, 13 (68.4%) were already primarily identified by CT scan in the emergency room, while 6 (31.6%) were depicted only by elective MRI evaluation. Therefore, we can state that MDCT has a sensitivity of 68%, a specificity of 100%, a negative predictive value of 50%, a positive predictive value of 100%, and an accuracy of 76% (Table 2).

**Table 2.** Contingency table of the MDCT results on 25 patients.

|  | Disease Present | Disease Absent |
|:---:|:---:|:---:|
| **MDCT Positive** | 13 | 0 |
| **MDCT Negative** | 6 | 6 |

Of the 19 affected patients, we analyzed radiologically a total of 22 vertebral segments that presented MRI alterations:

Of these, 4 (18.2%) were in the cervical region, 9 (40.9%) were at the thoracic level, and 9 (40.9%) were in the lumbar region. A total of 89% of patients with acute spondylodiscitis had only one involved segment; 11% of patients had two involved segments.

We evaluated the presence of MRI findings and their frequency, taking into account the distribution of segmental affections. The frequency of radiological findings on MRI was the following: edema-related changes in 22/22 cases (100%), erosive bone changes in 9/22 cases (40.9%), inflammatory tissue in the paravertebral or epidural space in 21/22 cases (95.5%), and pathological enhancement in the vertebral body or in the intervertebral disc in 22/22 cases (100%) (Figure 2, Table 3).

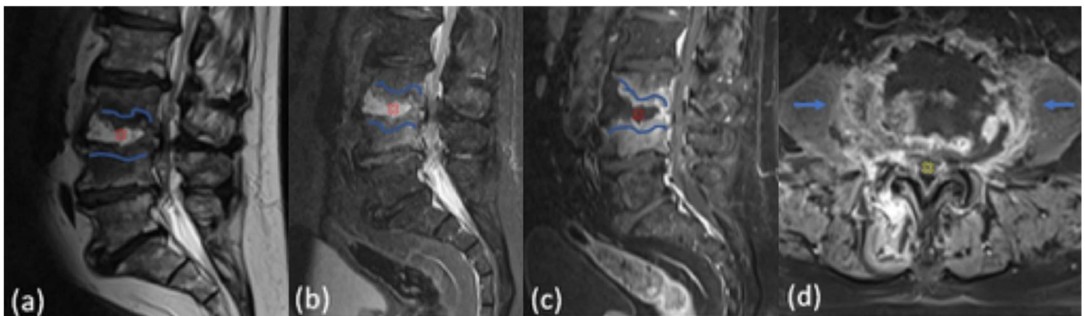

**Figure 2.** MRI TSE T2-weighted (**a**), STIR (**b**), and Dixon T1 CE+ (**c**) sequences on sagittal planes, and T1-weighted CE+ sequence on axial planes (**d**) show: subchondral bone erosions and bone edema at the end plates at the level L3–L4 ((**a–c**), curvilinear blue lines); significant edema of the vertebral disc as "hot disc" ((**a–c**), red point) with intense bone inhomogeneous enhancement of end plates; inhomogeneous enhancement in the adjacent soft tissues ((**d**), blue arrow) and epidural abscess ((**d**), yellow point).

We then evaluated the presence of MDCT findings and their frequency, taking into account the distribution of segmental affections. Of the 13 affected patients, we analyzed radiologically a total of 15 vertebral segments that presented MRI alterations: 3 (20%) were in the cervical region, 6 (40%) were at the thoracic level, and 6 (40%) were in the lumbar region. A total of 86.7% patients with acute spondylodiscitis had only one involved segment; 13.3% patients had two involved segments.

**Table 3.** Diagnostic findings of spondylodiscitis on MRI in 22 segments (19 patients).

| Finding | Location | Number |
|---|---|---|
| edema-related changes | vertebral body | 22 (100%) |
| erosive bone changes | vertebral body | 9 (40.9%) |
| inflammatory tissue | paravertebral and or epidural space | 21(95.5%) |
| contrast enhancement | vertebral body and or intervertebral disc | 22 (100%) |

The frequency of radiological findings on MDCT was the following: erosive bone changes in 13/15 cases (86%), inflammatory tissue in the paravertebral or epidural space in 14/15 cases (94%), and pathological enhancement in the vertebral body or in intervertebral disc in 100% of cases that underwent a MDCT with contrast media (9 patients) (Figure 3, Table 4).

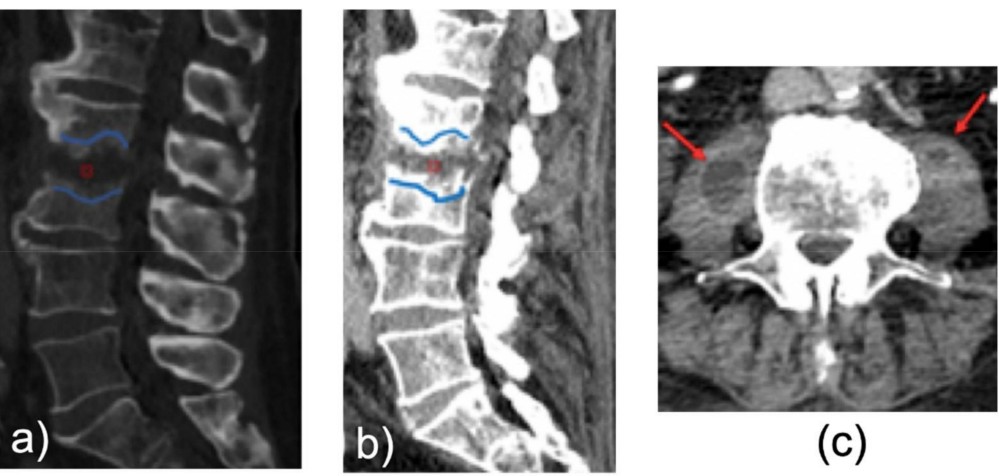

**Figure 3.** Sagittal CECT reconstruction in bone (**a**) and soft tissue (**b**) window, and axial CECT reconstruction in soft tissue window (**c**) show in L2-L3 level: moderate subchondral bone erosions of the end plates ((**a**,**b**), blue curvilinear lines) and strong enhancement of the subchondral bone of the end plates ((**b**), blue curvilinear lines), without significant densitometry changes in the disc ((**a**,**b**), red point) but with evident muscle abscesses of the ileus psoas muscles, especially on the right ((**c**), red arrows). CECT was suspicious for spondylodiscitis, confirmed by positive blood cultures.

**Table 4.** Diagnostic findings of spondylodiscitis on MDCT in 15 segments (13 patients).

| Finding | Location | Number |
|---|---|---|
| erosive bone changes | vertebral body | 13 (86%) |
| inflammatory tissue | paravertebral and or epidural space | 14 (94%) |
| contrast enhancement | vertebral body and or intervertebral disc | 9 CECT (100%) |

## 4. Discussion

In our study, MDCT achieved moderate sensitivity (68%), excellent specificity (100%), and satisfactory final accuracy (76%). In light of these results, we believe that CT can be used as a substitute or "to bridge" to MRI in cases where the patient does not tolerate MRI, or in cases where prompt action needs to be taken but there is no possibility of urgent MRI examination.

In addition, CT can detect some changes secondary to the infectious process more frequently than MRI. In fact, in detection of spondylodiscitis by MDCT, we analyzed some findings suggestive of pathology such as subchondral bone erosions and destruction of the endplates at a very early stage. Conventional radiology has low sensitivity and specificity in identifying and evaluating acute spondylodiscitis. In fact, the ability to assess bone matrix loss requires a 30% to 40% reduction in total bone matrix. This can take quite a long time, up to more than 2 weeks during the acute phase of infection [20]. Today, modern CT scanners allowing spiral volumetric acquisitions with thin-slice collimation and multiplane reconstructions are able to identify initial destructive endplate changes [21].

In addition, CT is very useful in identifying or excluding paravertebral or epidural abscesses that cannot be detected by radiographs. Extra-osseous abscesses can be better identified by intravenous administration of iodinated contrast medium, which allows visualization of the peripheral capsule of the abscess. We can measure the size and extent of the inflammatory abscess or granulation tissues, but not the inflammatory reaction of the bone marrow, unlike MRI [21].

Thus, CT can identify with high frequency every typical aspect of spondylodiscitis, including the presence of air in the disc, but it fails to assess direct signs of inflammation or infection. The presence of air is represented as a sign of empty disc space, indicative of very early infection by gas-forming bacteria or a fistula with the gastrointestinal or external tract. Occasionally, the disappearance of a previously visualized void sign can also be a clue to the presence of discitis. Here, then, is the reason for emphasizing the presence or disappearance of air in the context of an inflammatory process. Both the air void sign and destructive bone changes are often better appreciated with CT than with MRI [7].

Our retrospective study shows the main limitations of MDCT in the diagnosis of spondylodiscitis compared with MRI.

As reported by previous studies, MDCT does not identify edema changes [8,19,22].

On the other hand, the detection of an epidural abscess is important because, if not treated promptly, it can cause increased morbidity (e.g., persistent neurological deficits) and mortality [23], therefore, it is not possible to wait for an elective MRI exam in case of clinical suspicion of complications. The current debate on the indication for early surgical treatment of epidural abscesses is mainly based on the clinical impact that the resulting compression on nerve structures may cause [23–25], therefore, when available, it is always advisable to perform a CT scan.

Few scientific studies about MDCT have achieved a higher sensitivity than ours [14,15,19]. However, it seems unlikely that modern MDCT has lower sensitivity than older CT scanners in identifying and evaluating spondylodiscitis [26]. More likely, we can explain the lower diagnostic sensitivity of MDCT in our study by the retrospective design conducted on a sample of patients with heterogeneous CT indications in relation to other differential diagnoses. Another explanation could come from the introduction of more powerful MRI scanners and the development of dedicated spinal coils, together with increasingly high-performance fat-saturated imaging techniques that induce increased spatial resolution and contrast. Therefore, it is possible to identify the most minimal signal changes, such as bone marrow edema, in the earliest stages of acute spondylodiscitis. Consequently, if MRI, which has become increasingly sensitive diagnostically in recent years due to the described technological development, is used as the reference standard, this results in a more pronounced reduction in the sensitivity of MDCT in contemporary or more recent studies than in those of the past. Our study shows that the value of MDCT in the diagnosis of acute spondylodiscitis lies in its high specificity and high positive predictive value, especially if paravertebral abscesses are observed. Therefore, the presence of paravertebral abscesses on examination with MDCT, even in the absence of bone erosion, is a radiological finding that is highly indicative and predictive of acute spondylodiscitis. In our study, we report a much higher detection rate of paravertebral abscesses (94%) than the 69% rate reported in a previous study using a single-slice CT scanner [15], most likely due to the possibility of higher spatial and contrast resolution and the availability of multiplanar

reconstructions that can be easily achieved with modern MDCT scanners. One diagnostic technology that can improve the diagnostic sensitivity of CT in identifying the initial signs of spondylodiscitis is dual-energy CT (DECT). Dual-energy CT can help differentiate materials based on their different X-ray absorption characteristics, highlighting the different behavior of materials with different atomic numbers [27]. DECT has excellent diagnostic accuracy for detecting bone marrow edema, comparable to MRI. The utility of these recent possibilities is greatest where access to MRI is still very difficult [28]. However, this innovative CT technology is not yet widely available, especially in emergency departments, unlike MDCT.

*Limitations*

The main disadvantages of our study are the rather low study population. In addition, only a few patients underwent CT examination with contrast media, although it is known that the use of contrast media increases the diagnostic sensitivity of MDCT in infectious diseases because it allows better identification of the presence of an infectious focus, even at a distance [29,30].

## 5. Conclusions

Our study confirms the limitations of MDCT in identifying early radiological signs of acute spondylodiscitis compared with the high sensitivity of MRI. However, MDCT can indicate the diagnosis of this infectious disease condition with high certainty in relation to its high specificity, especially when paravertebral abscesses are found. MRI is not widely available in emergency departments, unlike MDCT, which allows for a valid comprehensive assessment of the patient regarding the more serious complications of the disease. In this regard, it is emphasized that although it is appropriate to supplement the CT examination with MRI, CT allows diagnoses of clinical pictures in the acute setting that require immediate intervention. Here, therefore, if spondylodiscitis is suspected, it is appropriate in the absence of rapid MRI to submit the patient to a preliminary CT scan. The scientific search for CT signs suggestive of acute spondylodiscitis with MDCT remains an interesting and valuable field of applied clinical diagnostic research. The management of patients with suspected spondylodiscitis in light of these results could be changed. In particular, following an urgent negative CT exam, we could not require an urgent confirmatory MRI. Excluding medical urgency (stage of complications), we could require an elective MRI examination in the short term. If the CT exam is positive, we could not perform an MRI evaluation. Therefore, further future studies, especially prospective studies, would be needed to confirm our findings.

**Author Contributions:** Conceptualization, A.N., F.S., S.T., M.L., M.T., L.G., and V.D.; Methodology, A.N., F.S., S.T., M.L., M.T., and L.G.; Software, A.N., F.S., M.T., and L.G.; Validation, A.N., F.S., S.T., M.L., M.T., L.G., and M.L.; Formal analysis, A.N., and F.S.; Investigation, A.N., F.S., S.T., M.L., M.T., L.G., M.L., M.G.C., A.V., F.F., G.P., O.V., M.G., F.D.S., M.C., R.C., C.R., and P.P.S.; Data curation, A.N., F.S., S.T., M.L., M.T., L.G., and M.L.; Writing—original draft preparation, A.N., M.T., and L.G.; Writing—review and editing, A.N., F.S., S.T., M.L., M.T., L.G., and M.L.; Visualization, A.N., F.S., S.T., M.L., M.T., L.G., and M.L.; Supervision, A.N., F.C., F.T., and V.D. Software: V.P., Data curation: C.S., M.I., G.T., P.P.S., L.D.G. Formal analysis: M.I. Supervision: G.T. Investigation: P.P.S., L.D.G. All authors have read and agreed to the published version of the manuscript.

**Funding:** This research received no external funding.

**Institutional Review Board Statement:** No need to request ethical review and approval for this study because it is based on the retrospective analysis of results from clinical radiological care activities.

**Informed Consent Statement:** Informed consent was obtained from all subjects involved in the study.

**Data Availability Statement:** The data presented in this study are available within the presented and described article. Further data can be requested from the corresponding author.

**Conflicts of Interest:** The authors declare no conflict of interest.

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
