# Peer review of "The Diagnostic Performance of Multi-Detector Computed Tomography (MDCT) in Depiction of Acute Spondylodiscitis in an Emergency Department"

_tomography, doi:10.3390/tomography8040160_

Round 1

Reviewer 1 Report

The authors describe the method of multi-detector computed tomography (MDCT) and compare its effectiveness compared with the MRI. Here are some comments/ suggestion to improve the clarity of the paper. As I have reported to the Editor, looks to me that this work, even if important in perspective, looks without a clear strong conclusion.

line 46: what ED means? Emergency Department? Not obvious. The authors explicitly report the acronym.

line 67: "Emergency situations"

line 82 : Is not clear to me why of the large sample of 162 patients, you included only the subsample of  patients that did the MDCT before MRI. The subsample of 137 patients did the MDCT after the MRI? In case not, you should just tell that the sample of patients who underwent both MRI and MDCT was 25.

line 88-89 even if self explanatory there is no caption in the flow diagram.

line 134 Is not clear to me what the percentage mean. I understand from these numbers that 19 patients out of 25 experienced bacteria, but what 1% by Candida mean? 1% is 0.19 patients. The authors could clarify if the percentage means theat all 19 patients have all bacteria and those are the average percentage. 

line 148-154:  the percentage relative at the different locations are calculated as an average of  all 19 patients?

line 206:  "emphasizing the presence"

Author Response

Reviewer #1

The authors describe the method of multi-detector computed tomography (MDCT) and compare its effectiveness compared with the MRI. Here are some comments/ suggestion to improve the clarity of the paper. As I have reported to the Editor, looks to me that this work, even if important in perspective, looks without a clear strong conclusion.

Authors' reply: We thank the Referee for the valuable comments but we would like to clarify a concept about the conclusions. In our opinion, this study presents a really important perspective because it clarifies the effectiveness of CT in radiological casees that require immediate intervention. Therefore the radiologist, faced with a negative CT finding, can suggest an MRI in election (not in urgency) with relative serenity and without therapeutic delays.

line 46: what ED means? Emergency Department? Not obvious. The authors explicitly report the acronym.

Authors' reply: We thank the Referee for the valuable suggestion, and we have modified the Manuscript accordingly

line 67: "Emergency situations"

Authors' reply: We thank the Referee for the valuable suggestion, and we have modified the Manuscript accordingly

line 82 : Is not clear to me why of the large sample of 162 patients, you included only the subsample of  patients that did the MDCT before MRI. The subsample of 137 patients did the MDCT after the MRI? In case not, you should just tell that the sample of patients who underwent both MRI and MDCT was 25.

Authors' reply: We thank the Referee to express this right doubt and let's try to clarify it. Our study takes into consideration the neuroradiological characteristics indicative of acute spondylodiscitis (lines 113-121) reported by our neuroradiologists in their structured reports drawn up at the time of the first diagnosis. The inclusion of patients with MRI prior to CT could have overestimated the accuracy of the CT as the neuroradiologist was influenced by the first exam (MRI). We added this explanationin diagram’s captions.

line 88-89 even if self explanatory there is no caption in the flow diagram.

Authors' reply: We thank the Referee and we added diagram’s captions.

line 134 Is not clear to me what the percentage mean. I understand from these numbers that 19 patients out of 25 experienced bacteria, but what 1% by Candida mean? 1% is 0.19 patients. The authors could clarify if the percentage means theat all 19 patients have all bacteria and those are the average percentage. 

Authors' reply: We thank the Referee and, apologyzing for the mistake, we have corrected the Manuscript accordingly

line 148-154:  the percentage relative at the different locations are calculated as an average of  all 19 patients?

Authors' reply: We thanks the Referee for the question. We calculated the percentage on the total of vertebral segments involved.

line 206:  "emphasizing the presence"

Authors' reply: We thank the Referee for the valuable suggestion, and we have modified the Manuscript accordingly

Reviewer 2 Report

The diagnostic performance of multi-detector computed tomography (MDCT) in depiction of acute spondylodiscitis in an emergency department.

The authors studied the possibility of using multidetector computed tomography (MDCT) as an alternative technique to detect spondylodiscitis pathology during emergency diagnostic in human subjects. They evaluated 25 patients using both MDCT and the gold standard (magnetic resonance imaging MRI). The authors reported a modest sensitivity of 68% using MDCT for the prediction of spondylodiscitis pathology in an emergency situation.

Strengths and weaknesses

The studied pathology is not common, and the results confirmed the MRI as the gold standard.

The low number of patients negatively affects the quality of the statistical results.

Main concerns

The experimental design is weak.

Institutional Review Board should approve the use of medical records for this retrospective research.

The medical image acquisition process is not standard for all patients. The results could be affected by the absence of a contrast agent in the patients.

Minor concerns

Comparison by gender is possible.

The subjects without a contrast agent should be excluded from the study because the process of obtaining images was different.

The tests were not performed using a standard procedure including the same operators.

The image analysis must be performed guaranteeing a blinding process.

Author Response

Reviewer #2

The diagnostic performance of multi-detector computed tomography (MDCT) in depiction of acute spondylodiscitis in an emergency department.

The authors studied the possibility of using multidetector computed tomography (MDCT) as an alternative technique to detect spondylodiscitis pathology during emergency diagnostic in human subjects. They evaluated 25 patients using both MDCT and the gold standard (magnetic resonance imaging MRI). The authors reported a modest sensitivity of 68% using MDCT for the prediction of spondylodiscitis pathology in an emergency situation.

Strengths and weaknesses

The studied pathology is not common, and the results confirmed the MRI as the gold standard.

The low number of patients negatively affects the quality of the statistical results.

Main concerns

The experimental design is weak.

Institutional Review Board should approve the use of medical records for this retrospective research.

The medical image acquisition process is not standard for all patients. The results could be affected by the absence of a contrast agent in the patients.

Minor concerns

Comparison by gender is possible.

The subjects without a contrast agent should be excluded from the study because the process of obtaining images was different.

The tests were not performed using a standard procedure including the same operators.

The image analysis must be performed guaranteeing a blinding process.

We thank the Referee for the avaluable comments.

We would like to clarify some concepts, as also done in response to the other reviewers:

- The study may seem weak for the small couple of patients but expanding the evaluation to the other subjects (patients who had MRI before CT) the results do not change;

- The institutional review committee approved the use of medical records for this research; all patients undergoing diagnostic tests at our institute sign an informed consent regarding the use of the data for scientific purposes;

- The acquisition process for CT scans of the spine is standardized for all patients; the only decision is on the use of the contrast medium. That said, the non-contrast evaluations were unaffected by the observation of the contrasted images. The evaluation of radiological characteristics in images with contrast medium was only subsequent to those without contrast medium;

- The diversity of operators should be overcome in an institution with radiology experts reporting by means of structured reports.

Reviewer 3 Report

The authors compare the diagnostic performance of multi-detector computed tomography (MDCT) in depiction of acute spondylodiscitis to magnetic resonance imaging (MRI) in an emergency department.

Overall, the study is well designed and also well performed and complemented with nice Figures. However, the study included only 25 patients, and MDCT identified only 13 patients of 19 with spondzylodiscitis confirmed with MRI.. This sensitivity was found to be insufficient, and the diagnosis based on the MSCT correctly identified only the late complications of spondylodiscitis, such as paravertebrall abscesses and endplate damage. These limitations should be stated, and the conclusions should be derrived with these in mind, otherwise, one could find it as a great tool, and miss a third of patients.

Please also indicate how many of 162 suspected patients ended up with a spondylodiscitis diagnosis based on the MRI examination, and correlate positive/negative predictive values with the results of MDCT.

Since this is a retrospective study, did the present management change in your hospital, do you nowdays use MDCT, and when possitive, dont use MRI, or you still use MRI to confirm, despite 100% specificity of MDCT? Al

Author Response

Reviewer #3

The authors compare the diagnostic performance of multi-detector computed tomography (MDCT) in depiction of acute spondylodiscitis to magnetic resonance imaging (MRI) in an emergency department.

Overall, the study is well designed and also well performed and complemented with nice Figures. However, the study included only 25 patients, and MDCT identified only 13 patients of 19 with spondzylodiscitis confirmed with MRI.. This sensitivity was found to be insufficient, and the diagnosis based on the MSCT correctly identified only the late complications of spondylodiscitis, such as paravertebrall abscesses and endplate damage. These limitations should be stated, and the conclusions should be derrived with these in mind, otherwise, one could find it as a great tool, and miss a third of patients.

Authors' reply: We thank the Referee for the valuable comment but the aim of our study is not to demonstrate a superiority of CT over MRI nor a possible substitutability. The aim of our study is clarifies the effectiveness of CT in radiological casees that require immediate intervention (stage of complications). Therefore the radiologist, faced with a negative CT finding, can suggest an MRI in election (not in urgency) with relative serenity and without therapeutic delays.

Please also indicate how many of 162 suspected patients ended up with a spondylodiscitis diagnosis based on the MRI examination, and correlate positive/negative predictive values with the results of MDCT.

Authors' reply: Of 162 suspected patients, 133 (82.1%) was confirmed by MRI scan. With MRI as the gold standard, all 133 positive patients were considered ill even with negative blood cultures (47/133; 35.3%). In this regard, we did not consider making comparisons with predictive values between CT and MRI. Furthermore, in light of the reviewers' comments, we also examined patients with CT after MRI, blinded, and we did not find a significant change in the accuracy of the MDCT.

Since this is a retrospective study, did the present management change in your hospital, do you nowdays use MDCT, and when possitive, dont use MRI, or you still use MRI to confirm, despite 100% specificity of MDCT?

Authors' reply: We thank the Referee for the valuable question. The management of patients with suspected spondylodiscitis in light of these results has changed. Currently, following an urgent negative CT exam, we do not require an urgent confirmatory MRI. Excluding medical urgency (stage of complications) we require an elective MRI examination in the short term. If CT exam is positive we don’t perform a MRI evaluation.

Reviewer 4 Report

1.      Introduction. Line 58. “The etiology is an infectious”. Should be rewritten as “It has an infectious etiology” or “It is provoked by an infection” or something similar.

2.      Introduction. Line 58. Correct is originate, it is a verb.

3.      Introduction. Line 62. Clinic reads ambiguous. Correct is “clinical picture” or “clinical symptomatology”

4.      Introduction. Line 62. “Fever and illness”. What does here “illness” mean?

5.      Introduction. Line 69. Correct is “it allows to identify”

6.      Introduction. Lines 70 - 71. Revise “clinic” as above.

7.      Methods. Line 83. Correct is average age. Also, to the age (63.18 years), standard deviation should be added.

8.      Methods. First, the CONSORT diagram is used for RCTs and not for descriptive studies. Second, this figure is not relevant, since does not provide any additional information to the data in text.

9.      Methods. Line 94. “We conducted MRI exams” should be rewritten to “MRI exams were conducted..”, since it is retrospective.

10.  Methods. The authors write that “Diagnosis were further confirmed by positive blood cultures” and at the same time in Introduction they state that “blood cultures are often negative”. So, in how many patients the blood cultures were positive?

11.  Methods. What does “common software” mean? Please state concretely.

12.  Methods. Line 131. Correct is available online.

13.  Results. Line 145. Correct is lumbar region, not district (it is geographical term).

14.  Results. Figure 1. The level (L3 - 4) of observed changes should be indicated.

15.  Abstract. Line 145. “Thanks its high”. Correct is “given to its high” or “due to its”.

16.  Abstract. Line 33. “but scant literature is produced” correct is “scarce literature is available or exists” or similar.

17.  Abstract. Line 34. MDCT is not defined.

18.  Abstract. Line 43. Value is written 2 times.

Author Response

Reviewer #4

  1. Line 58. “The etiology is an infectious”. Should be rewritten as “It has an infectious etiology” or “It is provoked by an infection” or something similar.

We thank the Referee for the valuable suggestion, and we have modified the Manuscript accordingly

  1. Introduction. Line 58. Correct is originate, it is a verb.

We thank the Referee for the valuable suggestion, and we have modified the Manuscript accordingly

  1. Introduction. Line 62. Clinic reads ambiguous. Correct is “clinical picture” or “clinical symptomatology”

We thank the Referee for the valuable suggestion, and we have modified the Manuscript accordingly

  1. Introduction. Line 62. “Fever and illness”. What does here “illness” mean?

We thank the Referee for the valuable suggestion, and we have modified illness in fatigue

  1. Introduction. Line 69. Correct is “it allows to identify”

We thank the Referee for the valuable suggestion, and we have modified the Manuscript accordingly

  1. Introduction. Lines 70 - 71. Revise “clinic” as above.

We thank the Referee for the valuable suggestion, and we have modified the Manuscript accordingly

  1. Methods. Line 83. Correct is average age. Also, to the age (63.18 years), standard deviation should be added.

We thank the Referee for the valuable suggestion, and we have modified the Manuscript accordingly

  1. Methods. First, the CONSORT diagram is used for RCTs and not for descriptive studies. Second, this figure is not relevant, since does not provide any additional information to the data in text.

We thank the Referee for the valuable comment but we inserted the CONSORT diagram in the Manuscript under the suggestion of another Referee

  1. Methods. Line 94. “We conducted MRI exams” should be rewritten to “MRI exams were conducted..”, since it is retrospective.

We thank the Referee for the valuable suggestion, and we have modified the Manuscript accordingly

  1. Methods. The authors write that “Diagnosis were further confirmed by positive blood cultures” and at the same time in Introduction they state that “blood cultures are often negative”. So, in how many patients the blood cultures were positive?

      We thank the Referee for the valuable comment  but even if in letterature the blood cultures are often negative, in our study all patient with positive MRI have a positive bloiid cultures.

  1. Methods. What does “common software” mean? Please state concretely.

We thank the Referee for the valuable suggestion, and we have modified the Manuscript accordingly

  1. Methods. Line 131. Correct is available online.

We thank the Referee for the valuable suggestion, and we have modified the Manuscript accordingly

  1. Results. Line 145. Correct is lumbar region, not district (it is geographical term).

We thank the Referee for the valuable suggestion, and we have modified the Manuscript accordingly

  1. Results. Figure 1. The level (L3 - 4) of observed changes should be indicated.

We thank the Referee for the valuable suggestion, and we have modified the Manuscript accordingly

  1. Abstract. Line 145. “Thanks its high”. Correct is “given to its high” or “due to its”.

We thank the Referee for the valuable suggestion, and we have modified the Manuscript accordingly

  1. Abstract. Line 33. “but scant literature is produced” correct is “scarce literature is available or exists” or similar.

We thank the Referee for the valuable suggestion, and we have modified the Manuscript accordingly

  1. Abstract. Line 34. MDCT is not defined.

We thank the Referee for the valuable suggestion, and we have modified the Manuscript accordingly

  1. Abstract. Line 43. Value is written 2 times.

We thank the Referee for the valuable suggestion, and we have modified the Manuscript accordingly

Round 2

Reviewer 1 Report

The authors have addressed all the questions and the comments I have done therefore I recommend this paper for publication

Author Response

We thank the Referee for the valuable comment. We are proud that he recommends our paper for publication.

Reviewer 2 Report

Although the authors replied to some of the observations, the changes needed are structural.

Author Response

We thank the referee for appreciating our changes and how we replied to some of the observation, even if he believes structural changes are needed.

This manuscript is a resubmission of an earlier submission. The following is a list of the peer review reports and author responses from that submission.

Round 1

Reviewer 1 Report

The authors submitted a paper in which they investigate the value multidetector computed tomography in the identification of spondylodiscitis. The aim of the study is interesting as it investigates the value of a widely available and prompt technique, CT, in the evaluation of a very serious condition and the paper would be a useful addition to the literature. However, the study is poorly developed, and the cohort of patients is relatively small to achieve sound conclusions. The paper should be extensively re-written and edited as the quality of the English language is inadequate. The design of the study should be extensively revised: the only relevant conclusion is that some radiological signs (i.e. paravertebral abscess) have high specificity; however, these signs are the typical manifestations of advanced disease and that is the reason for the high specificity (and this part should be properly discussed in the limitations paragraph). A study that aims to highlight the value of CT in the diagnosis of spondylodiscitis should focus on those signs that would address the patient to further exams or that would help the differential diagnosis. Advanced spondylodiscitis is not a difficult diagnosis from a clinical and radiological point-of-view, the most insidious cases are those in which the infection and the inflammatory process are still confined to the vertebral column: these cases are the real challenge, and these patients would benefit the most from early diagnosis and prompt therapy. CT has the advantage of being widely available and represents the first or second exam of choice in the evaluation of the vertebral column; in designing such a study you should ask the questions “what radiological sign at CT might suggest a concealed spondylodiscitis?” and “which signs are typical for other disease or specific for spondylodiscitis?”.

 Title: Adequate

 Abstract: Informative. In line with the main text.

 Keywords: appropriate.

 Informed Consent: appropriate

 Ethical Committee: appropriate

 References:

 - Satisfactory

 Figures and tables:

 - not satisfactory. You should provide high-quality images.

Minor Comments:

  • Line 46 “TCMD” instead of “MDCT”.
  • Line 65: “Inflammatory blood values could be elevated but blood 64 cultures are often negative”. Please add a reference(s).
  • Line 163: “We evaluated the presence of MRI and MDCT results and their frequency, taking into account the distribution of segmental affections, including all segments (22 segments out of 19 patients).” Repeated sentence.
  • Line 175-187: the figure caption is really hard-to-follow. I suggest to re-write it. You should analyze each figure part in the given order. Moreover, you should explain why did you consider this CT a false negative. In this case, I would have added spondylodiscitis to the differential. 
  • One of the advantages of CT on MRI is that it can depict air. The presence of air within the disk might be seen in the case of spondylodiscitis. Did you evaluate the presence of air within the disk?
  • Line 205: I would not consider 68% a high sensitivity.
  • Line 212: 81% sensitivity. This value is different from the one at line 205. How do you explain this?
  • Line 273-276: this paragraph is a bit off-topic.

Author Response

Reviewer 1:

The authors submitted a paper in which they investigate the value multidetector computed tomography in the identification of spondylodiscitis. The aim of the study is interesting as it investigates the value of a widely available and prompt technique, CT, in the evaluation of a very serious condition and the paper would be a useful addition to the literature. However, the study is poorly developed, and the cohort of patients is relatively small to achieve sound conclusions. The paper should be extensively re-written and edited as the quality of the English language is inadequate. The design of the study should be extensively revised: the only relevant conclusion is that some radiological signs (i.e. paravertebral abscess) have high specificity; however, these signs are the typical manifestations of advanced disease and that is the reason for the high specificity (and this part should be properly discussed in the limitations paragraph). A study that aims to highlight the value of CT in the diagnosis of spondylodiscitis should focus on those signs that would address the patient to further exams or that would help the differential diagnosis. Advanced spondylodiscitis is not a difficult diagnosis from a clinical and radiological point-of-view, the most insidious cases are those in which the infection and the inflammatory process are still confined to the vertebral column: these cases are the real challenge, and these patients would benefit the most from early diagnosis and prompt therapy. CT has the advantage of being widely available and represents the first or second exam of choice in the evaluation of the vertebral column; in designing such a study you should ask the questions “what radiological sign at CT might suggest a concealed spondylodiscitis?” and “which signs are typical for other disease or specific for spondylodiscitis?”.

Thank you very much for the constructive critical judgment of the paper. Therefore, based on your useful suggestions I have re-written and edited the text, adding new sentences in the discussion and limits sections. I wrote the conclusion clearly in a specific paragraph. I revised the bibliography by adding new bibliographic references.

I specified the limits of the study in the specific paragraph:

The main disadvantages of our study are represented by the rather low study population, which then led to a subdivision into two groups of even smaller patients for the comparison of the diagnostic accuracy between MDCT with contrast and MDCT without contrast, and from the retrospective design. It is well known and documented that the use of contrast media increases the diagnostic sensitivity of MDCT in infectious pathologies because they allow to better identify the presence of an infectious focus, even distant [30.Urban BA, Fishman EK. Tailored helical CT evaluation of acute abdomen. Radiogr Rev Publ Radiol Soc N Am Inc 2000; 20: 725–749; 31.Hammond NA, Nikolaidis P, Miller FH. Left lower-quadrant pain: guidelines from the American College of Radiology appropriateness criteria. Am Fam Physician 2010; 82: 766–770]. However, we were unable to demonstrate the diagnostic benefit of using the contrast agent, possibly due to the low number of patients in both groups. I also specified and reported in the same paragraph that that the presence of abscesses (i.e. paravertebral abscess) indicates an advanced stage of the disease and this explains its high specificity, while MDCT showed a reduced sensitivity in searching for the initial radiological signs of the presence of a suspected acute spondylodiscitis (i.e. bone marrow edema ), more precociously identifiable with MRI.

Moreover, I explained in the discussion paragraph that diagnostic technology that can improve the CT diagnostic sensitivity in the identification of the initial signs of spondylodiscitis is represented by Dual-Energy CT (DECT), different from MDCT used in our study. Dual-energy CT can help differentiate materials on the basis of their different x-ray absorption characteristics by highlighting the different behaviors of materials with different atomic numbers [28.Gosangi B, Mandell JC, Weaver MJ, Uyeda JW, Smith SE, Sodickson AD, Khurana B. Bone Marrow Edema at Dual-Energy CT: A Game Changer in the Emergency Department. Radiographics. 2020 May-Jun;40(3):859-874. doi: 10.1148/rg.2020190173. PMID: 32364883]. The DECT presents an excellent diagnostic accuracy to detect the bone marrow edema, comparable to MRI. The utility of these recent possibilities is maximum where access to MRI is still very difficult [29.Saba L, De Filippo M, Saba F, et al. Dual energy CT and research of the bone marrow edema: Comparison with MRI imaging. Indian J Radiol Imaging. 2019;29(4):386-390. doi:10.4103/ijri.IJRI_243_19]. However, this innovative CT technology is still not widely available, especially in emergency departments.

In conclusion:

Our study confirms the limitations of MDCT in identifying early radiological signs of acute spondylodiscitis compared to the high sensitivity of MRI. However, MDCT can indicate the diagnosis of this infectious pathological condition with great certainty in relation to its high specificity, especially when paravertebral abscess are found. MRI is not widely available in the emergency department and the scientific research for CT signs suggestive of acute spondylodiscitis with MDCT remains an interesting and valid field of applied clinical diagnostic research. Therefore, further future studies, especially prospective ones, would be needed to confirm our results.

 Title: Adequate

 Abstract: Informative. In line with the main text.

 Keywords: appropriate.

 Informed Consent: appropriate

 Ethical Committee: appropriate 

 References:

 - Satisfactory

 Figures and tables:

 - not satisfactory. You should provide high-quality images.

I redid the figures and the tables, reassembling, enlarging and dividing them.

The final result is that of greater simplification and greater contrast resolution.

I have appropriately added the captions in relation to the presentation of the figures in the text and I have simplified the objects inserted in the figures (i.e. arrows, curved lines, etc.) to better explain the radiological findings. 

Minor Comments:

  • Line 46 “TCMD” instead of “MDCT”.

Thank you for the the modification suggestion. In line 45 “TCMD” acronym was changed in “MDCT”.

  • Line 65: “Inflammatory blood values could be elevated but blood 64 cultures are often negative”. Please add a reference(s).

Thank you for pointing out this missing reference. I added the appropriate reference number  in the text in line 64 [5.Berbari EF, Kanj SS, Kowalski TJ et al. 2015 Infectious Diseases Society of America (IDSA) Clinical Practice Guidelines for the Di-agnosis and Treatment of Native Vertebral Osteomyelitis in Adults. Clin Infect Dis 2015; 61: e26–e46].

  • Line 163: “We evaluated the presence of MRI and MDCT results and their frequency, taking into account the distribution of segmental affections, including all segments (22 segments out of 19 patients).” Repeated sentence.

Thank you for the careful reading. I deleted the repetitive sentence.

  • Line 175-187: the figure caption is really hard-to-follow. I suggest to re-write it. You should analyze each figure part in the given order. Moreover, you should explain why did you consider this CT a false negative. In this case, I would have added spondylodiscitis to the differential. 

Thank you for the suggestion. I re-wrote it from the line 174 to the line 193. I analyzed each figure’s parts in the given order. I explained that the CT findings was not considered suggestive of acute spondylodiscitis, with moderate subchondral bone erosions of the end plates considered as arthrosic degenerative alterations, without significant enhancement of the subchondral bone of the end plates and without significant densitometric changes in the disc and in the perivertebral soft tissues or in the spinal canal. In conclusion I appropriately modified Figure 1 in two parts describing the MRI findings on one side (a,b,c,d) and the CT findings on the other (e,f,g).

  • One of the advantages of CT on MRI is that it can depict air. The presence of air within the disk might be seen in the case of spondylodiscitis. Did you evaluate the presence of air within the disk?

Thank you for the suggestion to improve the description of the CT findings. CT can identify the presence of air in the disk better than MRI thanks to the specific technique of this diagnostic method, based on the identification of the densitometric alteration, while MRI is very susceptible to artifacts induced by the presence of air, especially at the air/tissue interface. The presence of air is depicted as disc space vacuum sign. The vacuum sign is unlikely to be present in an infection; rare exceptions include an infection very early in its course, in case-reportable infections with gas-forming bacteria, or an infection due to fistula with the gastrointestinal tract [Diehn FE. Imaging of spine infection. Radiol Clin North Am 2012; 50: 777–798]. Occasionally disappearance of a previously visualized vacuum sign may be a clue to the presence of discitis [Diehn FE. Imaging of spine infection. Radiol Clin North Am 2012; 50: 777–798]. Both the vacuum sign and the lack of destructive changes are often better appreciated with CT than MRI [Diehn FE. Imaging of spine infection. Radiol Clin North Am 2012; 50: 777–798]. In conclusion, the presence of air represents a parameter that cannot be fully evaluated in MRI as in CT and is not very specific for acute spondylodiscitis. For this reason it was not a sign that we considered in our review of CT radiological signs of acute spondylodiscitis to compare with those equivalent identifiable on MRI, used as the gold standard of reference. I have added all these considerations previously described form the line 253 to the line 263 in the Discussion paragraph of the text.

  • Line 205: I would not consider 68% a high sensitivity.

Thank you for a more correct consideration of the diagnostic sensitivity value. I changed the the adjective “high” to “moderate” in line 224 of the revised manuscript and I added the adjective “high” in line 224 of the revised manuscript for the evalutaion of the diagnostic specificity value (100%).

  • Line 212: 81% sensitivity. This value is different from the one at line 205. How do you explain this?

Thank you for the careful reading of the text. I changed the value with the correct same previous value. Modification properly inserted in line 231 of the revised manuscript.

  • Line 273-276: this paragraph is a bit off-topic.

Thank you for the consideration of the appropriateness of the pragraph in the context of the text. I therefore proceeded to eliminate it from the manuscript.

Reviewer 2 Report

The authors proposed that MDCT can be a effective tool for detecting the acute spondylodiscitis in early stage. While it may be useful, some issues should still be emphasized.

Please consider shortening the title.

Please check the format requirement of abstract to see if the numbering structure is needed.

Please check the aberration word “TCMD” in L46.

Please specify the emergency situation or make it clearer.

Please rearrange the subfigures in figure 1 and 2, enlarge the figures, highlight the feature that the authors want the readers to see.

Please add the demographic table of subjects.

Please try to discuss more about the benefit and advantages of using MDCT method. The authors took too much paragraph to discuss about the disadvantages and limitations of this method, but the readers may be more interested in its significance.

Author Response

The authors proposed that MDCT can be a effective tool for detecting the acute spondylodiscitis in early stage. While it may be useful, some issues should still be emphasized.

Thank you for the consideration.

I emphasized these issues in the discussion section. In the evaluation of the MDCT images we took into account some findings suggestive of spondylodiscitis such as bony erosions and destructions of end plates. Subchondral bone erosions and destructions are also clearly visualized on plain films but much later than with MDCT. Traditional radiology has low sensitivity and specificity in the identification and evaluation of acute spondylodiscitis. The possibility of evaluating the loss of the bone matrix requires, in fact, a reduction from 30% to 40% of the total bone matrix. This can take quite a long time, up to more than 2 weeks during the acute phase of the infection, unlike MDCT. Our study demonstrates that the value of MDTC in the diagnosis of acute spondylodiscitis lies in its high specificity and high positive predictive value, especially if there is observation of paravertebral abscesses. They were identified in 92% of the vertebral metamers involved in spondylodiscitis. Therefore, the presence of paravertebral abscesses on examination with MDCT, even without bone erosion, is a highly indicative and predictive radi-ological finding of acute spondylodiscitis. In our study, we report a much higher detection rate of paravertebral abscesses (92%) than the rate of 69% reported in a previous study using a single-slice CT scanner [Wirtz DC, Genius I, Wildberger JE et al. Diagnostic and therapeutic management of lumbar and thoracic spondylodiscitis–an evaluation of 59 cases. Arch Orthop Trauma Surg 2000; 120: 245 25], very probably thanks to the possibility of a higher spatial and contrast resolution and to the availability of multiplanar reconstructions that can be easily obtained with modern MDCT scanners.

I summarized and clarified the fundamental role of MDCT in identifying this infectious pathological condition in the emergency department in the conclusion section.

MDCT can indicate the diagnosis of this infectious pathological condition with great certainty in relation to its high specificity, especially when paravertebral abscess are found. MRI is not widely available in the emergency department unlike MDCT, which represents a much more widespread and quick to use diagnostic method, allowing to have a valid global evaluation of the patient with the most serious complications of an acute spondylodiscitis such as abscesses in the peri-vertebral soft tissues. The scientific research for CT signs suggestive of acute spondylodiscitis with MDCT remains an interesting and valid field of applied clinical diagnostic research.

Please consider shortening the title.

Thank you for the invitation to simplify. I wrote a new title: “The diagnostic yield of multidetector computed tomography (MDCT) for the identification of acute spondylodiscitis in the emergency department”.

Please check the format requirement of abstract to see if the numbering structure is needed.

Thank you for the suggestion. I eliminated the numbering while maintaining the original structure, as described in the journal paper format.

Please check the aberration word “TCMD” in L46.

Thank you for the careful reading. I modified it in the line 45 of the revised manuscript.

Please specify the emergency situation or make it clearer.

Tank you for the suggestion. I specified in the text the use of MDCT in the emergency department.

Please rearrange the subfigures in figure 1 and 2, enlarge the figures, highlight the feature that the authors want the readers to see.

I re-did the figures, reassembling, enlarging and dividing them. The final result is that of greater simplification and greater contrast resolution. I have appropriately added the captions in relation to the presentation of the figures in the text and I have simplified the objects inserted in the figures (i.e. arrows, curved lines, etc.) to better explain the radiological findings. 

Please add the demographic table of subjects.

I added the required table in the text, showed below. I changed the numbering of the other tables.

Table 1. Demographic data of the subjects included in the study.

Patients

Sex

Age

1

M

60

2

F

65

3

M

61

4

M

63

5

M

63

6

F

70

7

M

57

8

M

58

9

M

57

10

F

65

11

M

63

12

F

70

13

M

61

14

M

69

15

F

64

16

M

60

17

F

71

18

M

67

19

M

65

20

F

70

21

F

64

22

M

65

23

F

71

24

F

70

25

M

69

Please try to discuss more about the benefit and advantages of using MDCT method. The authors took too much paragraph to discuss about the disadvantages and limitations of this method, but the readers may be more interested in its significance.

Thank you for the invitation to point out the advantages of the MDCT.

I described them in the conclusion section.

Our study confirms the limitations of MDCT in identifying early radiological signs of acute spondylodiscitis compared to the high sensitivity of MRI. However, MDCT can indicate the diagnosis of this infectious pathological condition with great certainty in relation to its high specificity, especially when paravertebral abscess are found. MRI is not widely available in the emergency department unlike MDCT, which represents a much-more widespread and quick to use diagnostic method, allowing to have a valid global evaluation of the patient with the most serious complications of an acute spondylodiscitis such as abscesses in the peri-vertebral soft tissues.

The scientific research for CT signs suggestive of acute spondylodiscitis with MDCT re-mains an interesting and valid field of applied clinical diagnostic research. Therefore, further future studies, especially prospective ones, would be needed to confirm our results.

Reviewer 3 Report

 General Comments:

Your paper is fairly well written with expected results. You suggested to use in cases of positive or negative cases with MDCT, then what is the role of MDCT? Is spondylodiscitis mostly acute? What finding of CT may not be detected on MRI? CT may be advantageous than MRI for the guided biopsy.

Specific Comments:

1. Page 2, line 77: ...accuration of TC....----->...accuracy of CT....

2. Is Pott's disease also included in spondylodiscitis?

3. Could SPECT/CT using MDP be better than CT or MRI for the valuation of spondylodiscitis?

Round 2

Reviewer 1 Report

Thank You for duly addressing my comments. I think that the Discussion of your paper has improved. Nonetheless, I do not think that this study has sufficiently scientific soundness to allow publication.

I suggest adding a CONSORT flow diagram to improve the comprehension of the study design. I found the study design really hard-to-follow and I had to draw a diagram myself to effectively comprehend the inclusion process. Results might be hard-to-comprehend in many sections: you should start with the number of patients that were included in the study, that is 25, I presume. MRI correctly identified spondylodiscitis in 19 out of 25 (76%). From here, some points should be clarified. The authors collected 22 vertebral segments with spondylodiscitis that underwent both CT and MRI. CT was positive in 15 of them. In the study, the authors analyzed only these fifteen patients but what about the other 7 patients that did not present any radiological sign of suspicion at CT? In these cases, the radiological sign that the authors analyzed are not consistent, therefore they essentially are false negative. These patients seem to be accounted in the computation of specificity and sensitivity but in Table 4 there are 25 events (patients or segments? You should indicate this in the caption). However, in Table 3 are considered only 15 patients.  I do not understand if these are serious flaws in the computation of the diagnostic performance of CT or some parts are missing. The Authors should re-write completely the results section because as it is now it would not be comprehensible to most readers. Moreover, you should involve a qualified statistician.
In my opinion, the sensitivity of CT is very low but the PPV is satisfactory (if the sign is present, the possibility that the patient really has the disease is high) and this is the real strength of your work. Moreover, the positive predictive value is mentioned only in the abstract, not in the Materials and Methods nor Results paragraphs.

Reviewer 2 Report

The authors have given a comprehensive responses to the reviewer, so the reviewer agreed to accept it.

Author Response

Thank you a lot for the postive review comment.